# Anticoagulant Rodenticides, Islands, and Animal Welfare Accountancy

**DOI:** 10.3390/ani9110919

**Published:** 2019-11-04

**Authors:** Penny Fisher, Karl J. Campbell, Gregg R. Howald, Bruce Warburton

**Affiliations:** 1Wildlife Ecology and Management Team, Landcare Research, Lincoln, Canterbury 7608, New Zealand; warburtonb@landcareresearch.co.nz; 2Island Conservation, Puerto Ayora, Galápagos Islands 200350, Ecuador; karl.campbell@islandconservation.org; 3School of Agriculture and Food Sciences, The University of Queensland, Gatton 4343, Australia; 4Island Conservation, Santa Cruz, CA 95060, USA; gregg.howald@islandconservation.org; 5Island Conservation, Vancouver, BC V6B, Canada

**Keywords:** rodenticide, anticoagulant, wildlife management, animal welfare, eradication, island, conservation

## Abstract

**Simple Summary:**

Anticoagulant rodenticides are a mainstay of rodent management in many domestic, municipal, agricultural, and conservation settings. Anticoagulant poisoning has poor welfare outcomes for mammals and birds and, worldwide, this means potentially very large numbers of animals are poisoned annually consequent (intended or not) to rodenticide use. Critical differences in use patterns of anticoagulants applied for ongoing rodent control, versus application for rodent eradication especially on islands, have clear implications for animal welfare costs measured as cumulative number of animals affected over time. Here we outline these differences and discuss how animal welfare considerations can be weighed in decisions to use anticoagulant rodenticides for island eradication attempts.

**Abstract:**

Anticoagulant rodenticides are used to manage rodents in domestic, municipal, agricultural, and conservation settings. In mammals and birds, anticoagulant poisoning causes extensive hemorrhagic disruption, with the primary cause of death being severe internal bleeding occurring over days. The combined severity and duration of these effects represent poor welfare outcomes for poisoned animals. Noting a lack of formal estimates of numbers of rodents and nontarget animals killed by anticoagulant poisoning, the ready availability and worldwide use of anticoagulants suggest that very large numbers of animals are affected globally. Scrutiny of this rodent control method from scientific, public, and regulatory perspectives is being driven largely by mounting evidence of environmental transfer of residual anticoagulants resulting in harmful exposure in wild or domestic animals, but there is also nascent concern for the welfare of targeted rodents. Rodent control incurs a cumulative ledger of animal welfare costs over time as target populations reduced by poisoning eventually recover to an extent requiring another reduction. This ‘rolling toll’ presents a critical contrast to the animal welfare accountancy ledger for eradication scenarios, where rodent populations can be completely removed by methods including anticoagulant use and then kept from coming back (e.g., on islands). Successful eradications remove any future need to control rodents and to incur the associated animal welfare costs.

## 1. Rodents, People, and Unwanted Consequences

Rats and mice (particularly *Rattus* spp. and *Mus musculus*) seem inevitably associated with human habitation. This association has largely negative outcomes for people, as rodents consume or spoil produce and stored resources, transmit epizootic diseases to people or domestic animals, and can damage infrastructure through gnawing or burrowing [1]. Historical reports of rodent control techniques reflect the duration of this commensal relationship [2], and rodent management remains routine worldwide particularly in and around buildings and in agricultural settings. In most countries, various products including traps, repellent devices, and baits containing rodenticide are marketed for household and industrial rodent control. Professional rodent management is a well-established service industry, implementing large-scale rodent baiting programs in municipal, industrial, and agricultural settings [3].

Ocean travel by humans has facilitated (and continues to facilitate) the introduction of commensal rodents to islands well beyond their natural dispersal capabilities [4]. Boats foundering near shore or merely visiting uninhabited islands have often inadvertently left behind a founder population of rodents. At least 85% of islands and island archipelagos around the world are estimated to have introduced populations of rodents [5]. The reproductive capacity and dietary and behavioral plasticity that allow rats and mice to coexist so successfully alongside human habitation [6] also make them formidable colonizers of insular island ecosystems.

As generalist omnivores [7], rats and mice readily adapt their foraging to utilize the most abundant and nutritious food resources which can include active predation of birds, reptiles, and invertebrates. For example, introduced mice have been confirmed to gnaw and eventually kill unfledged albatross chicks on some islands [8,9] and have been implicated in the ultimate cause of mortality in adult Laysan albatross on Midway Island [10]. For some animal species endemic to island ecosystems, naivety to competition or predation from introduced rodents has been a major factor in a significant decline in conservation status and, in some cases, extinction [11]. Introduced rodents are also engineers of island ecosystems, where their trophic role can influence and redistribute nutrient loads and alter vegetation regeneration, composition, and community structure [12]. Invasive rodents also dramatically alter the productivity and composition of surrounding marine ecosystems [13].

Regardless of the reason for wanting to reduce rodent impacts (e.g., biodiversity conservation, protection of crops, public health, or infrastructure), there are limited means available to achieve this effectively over large areas and for extended durations. Rodent control typically involves killing rodents rather than preventing them from breeding, excluding them from resources to be protected, or moving them elsewhere. Current lethal control tools are traps or rodenticides and, in particular, anticoagulant rodenticides are a mainstay of rodent management worldwide through a combination of efficacy, ease of use, relative safety for human operators, and cost-effectiveness [14]. Together, these attributes are typically considered when selecting lethal methods of rodent removal. Increasing awareness of, and concern for, environmental contamination and animal welfare are additional emerging influences on evaluating the acceptability of lethal methods of rodent removal in various contexts. Here we focus on animal welfare considerations with respect to the use of anticoagulant rodenticides and how these considerations differ in the contexts of rodent control versus rodent eradication.

## 2. Anticoagulant Rodenticides

Today, many people are familiar with the anticoagulant warfarin as an orally administered human medicine which acts as a blood thinner to prevent thrombosis. Warfarin (CAS number 81-81-2) was named for the Wisconsin Alumni Research Foundation (WARF), the organization that identified its anticoagulant properties and went on to develop it as both a rodenticide and therapeutic anticoagulant around 1950 [15]. Warfarin is one of a family of anticoagulant compounds based on a ‘coumarin’ chemical structure (Figure 1). The range of different anticoagulant compounds previously and currently used as rodenticides can be classified as indandiones or coumarins by ‘core’ chemical structure, and as first-generation (FGAR) or second-generation (SGAR) according to when they were first available as rodenticides (Figure 1).

The development of heritable resistance to FGARs in some Norway rat populations, particularly in the United Kingdom and Europe (e.g., [17]), saw declining efficacy of FGAR bait products, which prompted development and marketing of the more toxic SGARs in the 1970s [18]. The most potent SGAR compounds, particularly brodifacoum, remain effective against rodent populations that have developed resistance to other anticoagulants [19].

## 3. Toxic Action of Anticoagulants

The anticoagulants have a common mode of toxicity although there are differences between compounds in the degree of oral toxicity and in pharmacokinetics, conferred by variations in structure of functional groups [20]. In general, FGARs are most toxic when ingested as multiple, consecutive doses whereas the SGARs, particularly brodifacoum, are considered ‘single feed’ poisons for rodents because of their greater acute toxicity [21]. The generally lower oral toxicity of FGARs is attributed to a lower binding affinity for sites in liver microsomes [22,23]; in some rodent species, differences in toxicity of the same anticoagulant to males and females have been noted [21,24].

The ‘vitamin K cycle’, a set of sequential reactions that recycles vitamin K to its reduced form [25], drives the synthesis of several blood-clotting proteins within liver microsomes. Anticoagulants inhibit this cycle [26,27] through binding to a molecular target in the enzyme vitamin K epoxide reductase [28] which facilitates a critical step in the vitamin K cycle. Visible signs of anticoagulant poisoning in mammals are preceded by an asymptomatic (‘lag’) period, during which circulating levels of the vitamin K-dependent blood clotting factors become depleted [29] because they are not being renewed via a fully functional vitamin K cycle.

In rodents that ingest a lethal amount of anticoagulant bait, signs of poisoning typically become visible after a few days (Table 1). This delayed onset of poisoning is essential to the efficacy of anticoagulants, as rodents readily associate consumption of fast-acting rodenticides with subsequent negative physiological effects and quickly form bait aversions [30]. The absence of immediate toxic effects from ingesting anticoagulant bait means that rodents are more likely to continue consuming bait. Rats can consume well in excess of a lethal amount of anticoagulant bait before displaying visible effects of poisoning [31]. Typically, the first visible sign of anticoagulant poisoning in rodents is a significant decrease in food intake [21,32].

Illness in anticoagulant-poisoned rodents generally becomes outwardly visible within three to four days, with some variation (Table 1). Clinical signs also tend to last a further three to four days and times to death (as measured from when bait was consumed) are typically six to nine days after bait was first ingested (Table 1). While there is overall variation in time to first illness, duration of illness, and time to death, this does not appear to be related to the acute toxicity of the different anticoagulant compounds (i.e., rodents poisoned by the more toxic SGARs do not have shorter times to death).

## 4. Animal Welfare Outcomes of Anticoagulant Poisoning

In research aimed at evaluating animal welfare outcomes [39], interval-based behavioral observations were documented to describe the progression of brodifacoum poisoning in rats. Four days after ingesting a lethal amount of bait, rats showed reduced activity and a loss of appetite, and spent less time in a normal curled position when sleeping. This was accompanied by observations of affected rats standing in a hunched posture with lowered head, or lying down. One third of the rats underwent paresis and then paralysis two days before death. Rats that developed partial paralysis lay prostrate and were conscious for 11.4 hours, on average, before death. Clinical signs of anticoagulant poisoning in rats described by Mason and Littin [47] included external bleeding from orifices or wounds, pale extremities, and bloody diarrhea. Postmortem, rats in this study showed multiple hemorrhages in various sites including muscle, intestinal tract, joints, lungs, and viscera and the presence of free blood in body cavities and subcutaneous hematomas were common. Other studies report similar findings for rats, where anticoagulant poisoning results in gastrointestinal, orbital, intracranial, or other locations of hemorrhages described as ‘capable of producing severe pain’ [47]. Overall, anticoagulant poisoning in rodents potentially causes severe to extreme adverse physiological effects associated with impaired blood coagulation, occurring over a number of days. In an assessment of the animal welfare impacts of various control methods used to manage pest mammals in New Zealand [48], anticoagulants in general were ranked amongst those producing the most severe and prolonged poor animal welfare.

The welfare of rodents used in research or kept as companion animals is subject to formal regulation and community expectations (e.g., [49]) in many countries. Littin and colleagues [39] highlight a lack of similar regulation and societal regard for the welfare of rodents in situations where their free-living populations are subject to management as pests. Use of anticoagulants for rodent control has been framed as a ‘welfare paradox’ [50] because of adverse animal welfare outcomes of poisoning for very large numbers of animals as an ongoing, common and widely accepted occurrence worldwide. Interestingly, animal welfare concerns relating to anticoagulant use seem to have gained more recent prominence in the context of proposed eradication of rodents from islands [51] than as the result of decades of ongoing use for rodent control. Differences in how anticoagulants are used in the distinct scenarios of ‘rodent control’ or ‘rodent eradication’ confer different animal welfare outcomes for each scenario, as discussed further in Section 6 below.

## 5. Nontarget Effects of Anticoagulant Use

Use of anticoagulant rodenticides may lead to unintentional poisoning of nontarget wildlife or domestic pets (e.g., [52]). Primary poisoning occurs where nontarget animals ingest a harmful or lethal amount of bait—typically this involves omnivorous or herbivorous animals because most rodenticide bait formulations are cereal-based. Secondary exposure typically involves scavenging or predatory (including insectivorous) animals, when they consume tissues from other animals which carry residual concentrations of anticoagulant. The relatively higher toxicity and metabolic persistence of the SGARs, particularly brodifacoum [20], impart a higher secondary risk of mortality to nontarget animals in comparison to the FGARs [53].

Assuming similarly high welfare impacts of anticoagulant poisoning in mammals and birds, anticoagulant use accrues additional welfare costs through nontarget effects on domestic animals and wildlife. The potential for anticoagulant rodenticides to cause mortality in nontarget wildlife was known at least in the late 1970s [54]. In the mid–late 1990s, research investigating the connection between anticoagulant rodenticide use and poisoning mortality of nontarget wildlife was being published from around the world [55,56,57,58]. Research and monitoring has since demonstrated that globally, use of anticoagulant rodenticides is resulting in environmental transfer of residual concentrations of anticoagulants through trophic pathways and consequent mortality or morbidity in nontarget wildlife [59,60,61,62]. Identification of residual anticoagulant concentrations in freshwater and marine aquatic environments [63,64,65] indicates that unwanted effects of the use of anticoagulants for rodent management are not limited to terrestrial environments.

## 6. The Difference between Rodent Control and Rodent Eradication

In popular use, the terms ‘eradication’ and ‘control’ are often used interchangeably and considered to mean similar things. In more technical usage in the context of rodent management, ‘eradication’ describes the complete and permanent removal of a population, while ‘control’ involves repeated population reductions in what is essentially a sustainable harvest.

Typically, rodent control needs to be repeated regularly to reduce or maintain localized rodent populations to some nominal level at which they are no longer causing unacceptable damage, disease risk, or nuisance. In situations where eradication is not a realistic expectation or goal, rodent control leaves survivors, or the treated area is reinvaded, and the cycle repeats as the rodent population grows. Anticoagulant rodenticides are a mainstay of this approach to rodent management worldwide [14], however, this use pattern incurs a high, ongoing animal welfare cost in huge numbers of targeted rodents as well as large but unquantified numbers of nontarget animals.

Islands support ~40% of all threatened species, many threatened by introduced rodents [11]. Removal of rodents from islands is a demonstrated protection and recovery measure for threatened species and since the 1950s, the eradication of rodent populations from islands has become increasingly feasible and common, with nearly 600 successful eradications globally at a reported success rate of ~90% [66,67]. Success in eradicating invasive rodents from islands has, to date, been largely dependent on the use of the anticoagulant rodenticide brodifacoum [66,68,69] on uninhabited islands. Attention is increasingly upon the potential benefits of transferring this experience to rodent eradication from islands permanently inhabited by people, with acknowledged challenges in doing so [70,71].

Cowan and Warburton [72] highlighted the animal welfare implications of eradication failure where many thousands of the target animals are killed for, at best, a temporary reduction in their impacts—essentially the same critique that can be applied to rodent control. Importantly, Cowan and Warburton [72] did not consider animal welfare accountancy in situations of eradications on inhabited islands where anticoagulant rodenticides had been in historic or ongoing use for rodent control. In these situations, the cost of any proposed use of brodifacoum bait as a major component of a rodent eradication efforts needs to be weighed against the costs of any ongoing and future use of anticoagulants for rodent control. Successful rodent eradication would provide an effective bottom line for animal welfare costs, because the entire existing rodent population is removed in a single intervention and no further anticoagulant use would be required.

Opposition to proposed eradications of rodents on human-inhabited islands may have a strong reference to animal welfare [73] and environmental residue concerns [74]. Evaluation of such concerns, on an island-by-island basis, needs to be weighed against the history and current use patterns of anticoagulants on the island. Because people do not readily connect their personal, household, or farm activities directed to rodent control through anticoagulant use with wider environmental effects [75], we suggest it is important to initiate awareness of and effectively communicate these impacts even before the planning stages of an eradication. This will facilitate individual and community perspectives toward analyses of risks and benefits of anticoagulant use for rodent control as distinct from rodent eradication. This would be particularly important on islands where anticoagulants are already being used for rodent control but where eradication is proposed. In many such instances, community engagement is expected to raise concerns about the animal welfare impacts of poisoning, alongside concerns regarding the environmental residues and nontarget effects of using anticoagulant rodenticides.

## Figures and Tables

**Figure 1 animals-09-00919-f001:**
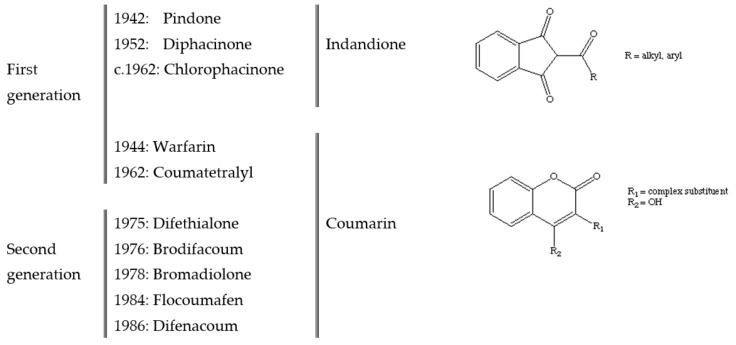
Date of development and use of first- and second-generation anticoagulant rodenticides, and their grouping by chemical structure [16]. c’—Circa abbreviated.

**Table 1 animals-09-00919-t001:** Times to death from anticoagulant poisoning reported in studies where captive rodents voluntarily ate food (bait) containing an anticoagulant rodenticide.

Anticoagulant/Rodent Species	Time to Observed Illness (days)	Time to Death (days)	Reference
**Warfarin**			
Laboratory *R. norvegicus*	mean 5.5	mean 6.0	[33]
Laboratory *R. norvegicus*	-	range 4.2–17	[34]
Laboratory *R. norvegicus*	-	mean 3.0 ± 0.45	[31]
Wild-caught *R. norvegicus*	-	mean 6.2–6.6 range 4.0–10.0	[35]
Wild-caught *R. rattus*	-	mean 7.1 range 3.0–13.0	[36]
Wild-caught *R. rattus*	-	mean (males) 7.9 range 4.0–13.0 mean (females) 7.0 range 2.0–13.0	[37]
Wild-caught *R. rattus*	-	means 7.0 and 7.5	[34]
Wild-caught *R. rattus*	-	mean 8.8 range 5.0–12.0	[35]
Wild-caught *Mus musculus*	-	range 3.0–30.0	[38]
**Coumatetralyl**			
Laboratory *R. norvegicus*	mean 1.2	mean 5.7	[33]
Laboratory *R. norvegicus*	-	mean 6.6 ± 0.5	[31]
Wild-caught *R. rattus*	-	mean 8.1 range 5.0–13.0	[36]
**Difenacoum**			
Laboratory *R. norvegicus*	mean 8.4	mean 11.0	[33]
**Brodifacoum**			
Laboratory *R. norvegicus*	mean 4.0	mean 13.3	[33]
Laboratory *R. norvegicus*	mean 3.0	mean 7.2 range 5.6-8.5	[39]
Laboratory *R. norvegicus*	-	mean 4.3 ± 0.5	[31]
Wild-caught *R. rattus*	-	mean 6.9 ± 1.9 range 3.0–13.0	[40]
Wild-caught *M. musculus*	-	mean 9.9 range 6.0–18.0	[41]
Wild-caught *M. musculus*	-	mean 9.0 ± 0.6 range 3.0–21.0	[42]
Wild-caught *M. musculus*	-	range 4.0–19.0	[43]
Wild-caught *M. musculus*		mean 5.5± 2.5 range 1.0–16.0 median 6.0	[44]
Wild-caught *M. musculus*		mean 7.3 ± 3.9 range 1.0–18.0	[40]
**Bromadiolone**			
Wild-caught *R. norvegicus*	-	range 4.0–6.2	[45]
Pindone			
Laboratory *R. norvegicus*	-	mean 4.0 ± 0.5	[31]
Wild-caught *R. rattus*	-	mean 7.9 range 3.0–19.0	[36]
Wild-caught *R. norvegicus*	-	means 4.2 and 5.0 range 4.0–9.0	[35]
Wild-caught *R. rattus*	-	means 8.7 and 8.7 range 4.0–13.0	[35]
Wild-caught *M. musculus*	-	range 4.0–6.0	[46]
**Diphacinone**			
Laboratory *R. norvegicus*	-	mean 2.9 ± 1.8	[31]
**Chlorophacinone**			
Wild-caught *R. rattus*	-	mean 9.7 range 4.0–19.0	[36]

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
