# Peer review of "Anticoagulant Rodenticides, Islands, and Animal Welfare Accountancy"

_animals, 2019, doi:10.3390/ani9110919_

Round 1

Reviewer 1 Report

This manuscript is original and provides useful information for various parties. It does a good job of clarifying the difference between sustained anticoagulant use versus its use for island rodent eradication. It is well organized and written. It also flows very well. There is good use of the scientific literature. The table and figure are appropriate. I have no editorial or content suggestions.

Author Response

No response needed

Reviewer 2 Report

Review for “Anticoagulant rodenticides, islands and animal welfare accountancy” by Fisher et al. (ms. animals-622786). This manuscript provides commentary on how anticoagulants impact animal welfare, both for target and nontarget species. The emphasis is on how island eradications can reduce animal welfare concerns over the long-term by requiring only a pulse application of anticoagulant to remove all island invaders, thereby eliminating the need for additional applications to continually reduce rodent numbers. As such, the number of animals impacted will be much lower long-term than if continual control efforts were needed. This manuscript was very well written, which is greatly appreciated. The manuscript also serves as a good review of material on several aspects pertaining to anticoagulant application, animal welfare, and nontarget risk. I only have a few minor suggestions to improve the manuscript. You can find these suggestions below.

Table 1: I really appreciate this Table and the amount of work that went into collecting all this data. One item that I do feel would increase its usefulness would be to add one additional column that would list sample sizes for the mean and range values. Would this be possible?

Lines 14, 25, and 173: You “estimate” that there are millions of domestic animals and wildlife that are poisoned annually. This number “may” be somewhat close, but we do not know this. Furthermore, you provide no information on how you estimated this value. I think it would be better to define this in a different capacity given that we really do not have any actual numbers to ascribe to this. For example, on Line 173, perhaps state: “…potentially affecting a substantial number of animals annually”. I know this may not have the impact that you prefer, but it is more defensible.

Line 207: I think that “as” should be “an”.

Author Response

Table 1: I really appreciate this Table and the amount of work that went into collecting all this data. One item that I do feel would increase its usefulness would be to add one additional column that would list sample sizes for the mean and range values. Would this be possible?

Unfortunately compiling the suggested additional information for Table 1 would substantially delay finalisation of the manuscript. The primary author (who compiled the table some years ago) does not have immediate access to the now-archived hard copies of some of the references cited, noting that many of these were hard to find and not readily available in electronic format. Even with immediate access to these references there would be significant time required to go back through each study in order to determine whether information about sample sizes could be readily added to each citation. The authors contend that adding information about sample sizes would be ‘nice to have’ rather than adding substantially to the main value of Table 1. The intention of Table 1 was to demonstrate the consistency in times to first signs and times to death reported across literature describing anticoagulant poisoning in rodents, despite differences in species and anticoagulant compound.

Lines 14, 25, and 173: You “estimate” that there are millions of domestic animals and wildlife that are poisoned annually. This number “may” be somewhat close, but we do not know this. Furthermore, you provide no information on how you estimated this value. I think it would be better to define this in a different capacity given that we really do not have any actual numbers to ascribe to this. For example, on Line 173, perhaps state: “…potentially affecting a substantial number of animals annually”. I know this may not have the impact that you prefer, but it is more defensible.

Lines 14, 25: amended to remove the estimate of millions, note the lack of formal estimates and make the statements about numbers affected less definitive. Line 173 have removed any reference to estimated numbers of affected animals.

Line 207: I think that “as” should be “an”.

Amended – thank you.

Reviewer 3 Report

The authors present a brief commentary piece on a topic that is important and potentially of wide interest, namely anti-coagulant rodenticide and it's use, ethics and impacts. I found myself a little unsure of the role of this contribution. It had no clearly stated aim or rationale or introduction and jumped into a situation summary of many aspects of rodenticides and their use. Given that there have been many thorough and recent reviews on this topic (see references provided below as examples), the role of this contribution needs to be more tightly defined. The authors are clearly highly experienced in this area and I have no doubt that they have much to offer, but at least the written piece presented lacked a clear explanation for the reader of the offering and it's key argument. I particularly liked where things were heading at the end with regards to conflict around island eradications, but little was offered to the reader in this space.

If the authors can focus on clearly defining the scope for this contribution, especially in a space where there is much recent literature, including better references to that literature and delivering a compelling take home message or some new ideas, then I feel this could be a valuable contribution. As it stands the message is unclear and the offering limited.

Specific comments:

Line 22-24 needs rewording due to grammatical disjunction. Try something like the following: "In mammals and birds, anticoagulant poisoning causes extensive hemorrhagic disruption, with the primary cause of death being severe internal bleeding occurring over days"

Line 25: "We estimate at least millions". Either cite some estimates or reword to something like "Potentially millions".

Line 58, comma after "omnivores"

Line 59 end needs a reference. The next sentences do mention vertebrate predation but not invertebrates or reptiles.

Line 154 on can I suggest the following reference as it is more current than. many referenced and seems to be the most recent and comprehensive review on wildlife impacts I have read: Lohr, M. T. and Davis, R.A. (2018). Anticoagulant rodenticide use, non-target impacts and regulation: a case study from Australia. Science of the Total Environment. 634: 1372-1384.

Can I also recommend this as another recent contribution: Nakayama, S., Morita, A., Ikenaka, Y., Mizukawa, H., & Ishizuka, M. (2019). A review: poisoning by anticoagulant rodenticides in non-target animals globally. The Journal of veterinary medical science, 81(2), 298–313. doi:10.1292/jvms.17-0717

Some further detail could be added in this section as it doesn't include very much information on wildlife impacts in the text.

Line 186 "Islands support 40% of threatened species" needs a reference. Also to the fact that many are threatened by rodents. Can I recommend the following citation: Doherty TS, Glen AS, Nimmo DG, Ritchie EG, Dickman CR. (2016) Invasive predators and global biodiversity loss. Proceedings of the National Academy of Sciences USA.

Author Response

The authors present a brief commentary piece on a topic that is important and potentially of wide interest, namely anti-coagulant rodenticide and it's use, ethics and impacts. I found myself a little unsure of the role of this contribution. It had no clearly stated aim or rationale or introduction and jumped into a situation summary of many aspects of rodenticides and their use. Given that there have been many thorough and recent reviews on this topic (see references provided below as examples), the role of this contribution needs to be more tightly defined. The authors are clearly highly experienced in this area and I have no doubt that they have much to offer, but at least the written piece presented lacked a clear explanation for the reader of the offering and it's key argument. I particularly liked where things were heading at the end with regards to conflict around island eradications, but little was offered to the reader in this space.

If the authors can focus on clearly defining the scope for this contribution, especially in a space where there is much recent literature, including better references to that literature and delivering a compelling take home message or some new ideas, then I feel this could be a valuable contribution. As it stands the message is unclear and the offering limited.

We agree with reviewer 3 that there is rapidly growing literature and ample recent publications describing the incidence and frequency of anticoagulant residues in wildlife and natural environments, and the mode of toxic action and pathology associated with anticoagulant exposure and poisoning. Hence this paper was not represented as a review of these themes any more than was necessary background for the “key argument” - being the animal welfare implications of different anticoagulant use patterns.

As indicated by reviewer 3, we have added some changes to try and make the “take home message” relating to animal welfare more apparent as a valuable contribution. At the end of different sections, we have added additional commentary to reinforce the animal welfare thread of discussion. See tracked text lines 78-83, lines 168-172 and lines 237-242.

Reviewer 3’s preferences for discussion of conflict around island eradications and for citing “better” recently-published references relating to anticoagulant residue profiles reported in wildlife are noted. We respectfully decline to accommodate these any further than the changes described here, as we feel this would otherwise require a substantial revision and change of title, and would also detract from the animal welfare focus intended to be in thematic keeping with this special issue of Animals.

Specific comments:

Line 22-24 needs rewording due to grammatical disjunction. Try something like the following: "In mammals and birds, anticoagulant poisoning causes extensive hemorrhagic disruption, with the primary cause of death being severe internal bleeding occurring over days"

Amended – thank you

Line 25: "We estimate at least millions". Either cite some estimates or reword to something like "Potentially millions".

See our response to Reviewer 2, who made a similar suggestion for revision.

Line 58, comma after "omnivores"

Added – thank you

Line 59 end needs a reference. The next sentences do mention vertebrate predation but not invertebrates or reptiles.

Reference citation added to the sentence in line 59: Landry Jr, S.O. The Rodentia as omnivores. The Quarterly Review of Biology 1970, 45, 351-372

Text in the following sentence amended to denote that the predation of albatross by mice was used as an example. Further review of rodents as predators of various other island fauna (reptiles also being vertebrates) was considered unnecessary for the main focus of this paper.

Line 154 on can I suggest the following reference as it is more current than. many referenced and seems to be the most recent and comprehensive review on wildlife impacts I have read: Lohr, M. T. and Davis, R.A. (2018). Anticoagulant rodenticide use, non-target impacts and regulation: a case study from Australia. Science of the Total Environment. 634: 1372-1384. Can I also recommend this as another recent contribution: Nakayama, S., Morita, A., Ikenaka, Y., Mizukawa, H., & Ishizuka, M. (2019). A review: poisoning by anticoagulant rodenticides in non-target animals globally. The Journal of veterinary medical science, 81(2), 298–313. doi:10.1292/jvms.17-0717

The original MS lines 165-168 cited two recent references (dated 2016 and 2018) with respect to current reporting of anticoagulant residues in wildlife and associated impacts:

Elliott, J.E.; Rattner, B.A.; Shore, R.F.; Van Den Brink, N.W. Paying the pipers: mitigating the impact of anticoagulant rodenticides on predators and scavengers. Bioscience 2016, 66, 401-407.

López-Perea, J.J.; Mateo, R. Secondary exposure to anticoagulant rodenticides and effects on predators. In Anticoagulant rodenticides and wildlife, Springer: 2018; pp. 159-193.

As suggested, we have added the Lohr & Davis 2018 and Nakayama et al. 2019 references to these existing citations in further support of this same statement, lines 180-183 in revised MS.

Some further detail could be added in this section as it doesn't include very much information on wildlife impacts in the text.

We decline to add further detail to this section as reviewer 3 suggests. This section was intentionally brief in seeking to articulate simply why non-target wildlife should be included in the count of animals potentially affected by anticoagulants, and hence in considerations of animal welfare. It was not within the scope of this paper to undertake another review of the impacts of anticoagulants on wildlife from an ecotoxicological or population perspective. As reviewer 3 points out, that is an area is being well covered in growing literature, and we have cited four chapters from a very recent book on this topic : Anticoagulant rodenticides and wildlife, Springer: 2018; pp. 159-193, for those readers interested in further investigation.

Line 186 "Islands support 40% of threatened species" needs a reference. Also to the fact that many are threatened by rodents. Can I recommend the following citation: Doherty TS, Glen AS, Nimmo DG, Ritchie EG, Dickman CR. (2016) Invasive predators and global biodiversity loss. Proceedings of the National Academy of Sciences USA.

This citation added line 198 – thank you.

Round 2

Reviewer 3 Report

The authors revisions have addressed my concerns and I thank them for their efforts. Only one remaining minor error:

Line 84: additional emerging influences evaluating the acceptability of is missing a noun. "influences when evaluating" or "on evaluating"?

Author Response

Line 84: have added "on" as a preposition - thank you for spotting the error